# How Job Characteristics Influence Healthcare Workers' Happiness: A Serial Mediation Path Based on Autonomous Motivation and Adaptive Performance

Ana Junça-Silva [1,2,*] and Catarina Menino [2]

1   Business Research Unit-IUL, ISCTE—Instituto Universitário de Lisboa, 1649-026 Lisboa, Portugal
2   Management Superior School, IPT—Instituto Politécnico de Tomar, 2300-313 Tomar, Portugal
*   Correspondence: ana_luisa_silva@iscte-iul.pt

**Abstract:** Self-determination theory suggests that motivation is multidimensional; as such, there are various dimensions ranging from autonomous (i.e., intrinsic) to more controlled forms (extrinsic) of motivation. While intrinsic motivation appears to be positively related to an individual's optimal functioning (e.g., happiness and performance), extrinsic motivation appears to be less beneficial. Furthermore, motivation is strongly determined by the context (e.g., job characteristics, such as autonomy). Although the relationship between job characteristics and workers' motivation has been demonstrated, how it impacts performance and happiness is still to be unpacked. Moreover, it is relevant to analyze such models within healthcare workers; their work is emotionally and psychologically demanding, hence, understanding what drives their intrinsic motivation is of crucial importance. Thereby, the aim of the study was to analyze the mediating role of intrinsic motivation and adaptive performance on the relationship between job characteristics and happiness. Based on the job characteristics model, we proposed a serial path from motivating job characteristics (autonomy, feedback, variety, meaning, and task identity) to healthcare workers' happiness via intrinsic motivation and adaptive performance, which was justified using the self-determination theory. We also argue that this path would not be significant for extrinsic motivation. We gathered data from 290 healthcare workers from a nursing home. The data were collected at three time points. The results support our hypotheses by demonstrating that all job motivating characteristics (autonomy, feedback, variety, meaning, and task identity) predicted healthcare workers' happiness by enhancing their intrinsic motivation and leading to better adaptive performances. The results are not significant for extrinsic motivation; that is, the serial mediating path was not significant when extrinsic motivation was analyzed. The findings highlight the need for managers to focus on work design, in a way to promote certain job motivating characteristics (e.g., autonomy), to improve healthcare workers' motivation, which leads them to achieve greater performances and, consequently, be happier. The study highlights that when healthcare workers have a job that provides them autonomy and regular feedback, with meaningful and varied tasks to which they feel a sense of identification, they tend to feel intrinsically motivated in their work, promoting higher adaptability to daily challenges, and, as a result, leaves them happier. The role of motivation and performance in the happiness of employees in a healthcare setting.

**Keywords:** intrinsic motivation; extrinsic motivation; job-characteristics model; performance; happiness

## 1. Introduction

Researchers have acknowledged the importance of understanding organizational characteristics that satisfy and motivate employees, because when they feel motivated, they are more productive and tend to be happier [1]. Motivation is the energy, direction, and persistence of behavior [2]; it is an essential component for organizations, as it influences individuals' work-related behavior [3]. Indeed, for workers to perform their

tasks effectively, they must feel motivated, because, with more will, they put more effort into their tasks, are more productive, and, thereby, achieve personal satisfaction [4]. The self-determination theory [5] suggests that motivation is determined by different types of goal-directed behavioral regulations that reflect psychological states. This theory further proposes that behavioral regulations, while being specific, are organized along a single continuum of self-determination [5,6]. That is, motivation is multidimensional and includes different dimensions, ranging from autonomous or intrinsic motivation to more controlled forms of motivation (e.g., external regulation). While autonomous motivation appears to be positively related to an individual's optimal functioning (e.g., well-being and performance), controlled motivation appears to be less beneficial [6].

Job characteristics are a key factor in understanding the relationship between motivation, performance, and happiness indicators [7]. The job characteristics model [8] proposes that certain job characteristics (task variety, significance, meaning, autonomy, and task feedback) are motivating in their nature and, as a result, lead to increased performance. Despite the several empirical demonstrations of the relationship between job characteristics, motivation, performance, and happiness, these have explored the isolated the impact of each job characteristic, associated with a single type of motivation, typically the autonomous one—intrinsic motivation [9]. However, given that motivation is multidimensional [10], and that different dimensions of motivation may have different situational predictors (e.g., job characteristics), it is important to analyze whether certain job motivating characteristics influence motivation (autonomous versus controlled) and lead to better performance and happiness [10]. This is important for the healthcare working context; healthcare employees deliver care and services to the sick and ailing. They include doctors and nurses or assistants, technicians, aides, or medical waste handlers [11]. Healthcare services are labor-intensive. Typically, they have to work in highly challenging and demanding working conditions [12] that, in the long run, not only leave them exhausted but also demotivated. Moreover, if one considers the COVID-19 pandemic crisis, which has been a period characterized by increased uncertainty, it highlights the need to understand what might improve these employees' motivation, performance, and happiness.

The main contribution of this study is to address each of these issues by (1) extending the job characteristics model to the healthcare context and (2) supposing that these professionals' happiness may be affected by a serial path that starts in their job characteristics (autonomy, feedback, variety, meaning, and task identity), which influence their intrinsic motivation and in turn promote higher adaptability. Since their working day is full of daily challenges and demands, we argue that this serial mediating path is not significant when extrinsic motivation is considered.

Hence, relying on the job characteristics model, we propose a model consisting of a motivational and behavioral path, based on a serial mediation—motivation and performance. The main research question was: what job characteristics would influence healthcare employees' intrinsic motivation that, in turn, make them more adaptable and result in increased happiness? Therefore, we aimed to analyze the path from job characteristics (autonomy, feedback, variety, meaning, and task identity) to happiness, via autonomous versus controlled motivation, and performance among healthcare workers.

This paper is divided into five sections. First, we describe the main theoretical assumptions that support the hypotheses and the serial mediation model. Next, we describe the methods section, followed by the hypotheses testing and its results. We finish with the discussion of results in light of the theoretical and former empirical evidence.

## 2. Theoretical Framework

### 2.1. The Relationship between Motivation and Happiness

The study of happiness appears associated with two major theoretical perspectives: hedonism and eudaimonism [13].

The hedonic approach emphasizes the subjective nature of happiness, considering individuals as the final "judges" of their life experiences [14]. The prevailing view of

hedonism operates on the premise that well-being is the experience of pleasure versus pain and adopts the concept of subjective well-being [15]. Subjective well-being is characterized by two components: (1) affective (higher frequency of positive emotions over negative ones) and (2) cognitive (evaluation that the individual makes about his/her life) [16]. Accordingly, a happy individual makes a positive evaluation of life and experiences positive emotions more often than negative ones [17]. In addition, according to Diener [18], individuals are happy when: (1) they often experience more positive than negative emotions; (2) are involved in interesting activities; (3) experience more pleasure than pain; and (4) are overall satisfied with their lives.

On the other hand, eudaimonism is considered by several authors [19] as a form of happiness that goes beyond obtaining pleasure alone and implies the development of the self to obtain happiness. For Ryan and Deci [15], the eudaimonic perspective focuses on meaning and self-fulfillment, defining well-being through the degree to which a person is fully functioning—psychological well-being. This perspective has been closely related to the concept of individuals' motivation, as proposed by the self-determination theory (SDT) [15]. Accordingly, individuals are motivated to satisfy three basic psychological needs: autonomy (having the independence to take decisions and control), competence (feeling competent regarding the environment surrounding), and relatedness (having positive relations with others) [15].

Motivation was defined as the energy, direction, and persistence of behavior [2]. It is a multidimensional concept [3] that regulates and sustains the individual's actions. Bergamini [20] argued that motivation is stimulated by two factors: internal and external. Internal factors are the intrinsic reasons, such as needs, aptitudes, interests, values, and abilities, through which the worker performs the tasks. External factors are environmental stimuli, that is, personal goals associated with rewards for a given effort, result, or behavior. The self-determination theory (SDT) [15] also proposed that motivation may be intrinsic or extrinsic. Indeed, the SDT offers a well-justified and empirically demonstrated theoretical framework [21]. It proposes that motivation is multidimensional because different dimensions of motivation have different underlying behavioral regulations, and as a consequence, different behaviors. Accordingly, motivation follows a continuum, ranging from more controlled forms of motivation (extrinsic motivation) to more autonomous forms (intrinsic motivation) [15]. This continuum ranges from motivation, external regulation, introjected regulation, identified regulation, integrated regulation, and intrinsic motivation.

Extrinsic motivation occurs when an individual performs a task for an instrumental reason, such as an external reward [22], and includes external (social and material), introjected, and identified regulation. On the other hand, intrinsic motivation occurs when an individual performs a task for the pleasure inherent in the activity itself. Thus, it is defined as an action for the simple satisfaction of performing it, and not for a consequence that arises from it [23]. When intrinsically motivated, the person is moved to act for fun or challenge, and not for external stimuli, such as pressures or rewards [15].

Most activities that people perform are not intrinsically motivated; that is, people need external stimuli to be motivated for certain tasks, i.e., they need extrinsic motivation [24]. Ryan and Deci [15] indeed suggested that behaviors arising from extrinsic motivation are not intrinsically interesting and, therefore, need external stimuli. This is more emphasized in the healthcare working setting. These contexts are psychologically demanding [12], which makes healthcare workers more vulnerable to psychological issues. Healthcare workers deliver care and services to the sick and ailing either directly as doctors and nurses or indirectly as assistants, technicians, aides, or medical waste handlers [11]. Most of these workers, regardless of their specialty, tend to experience decreases in motivation due to their psychologically demanding work, which undoubtedly changes the way they work, being unable to deliver the quality of care they normally would. Hence, for these workers, additional external stimuli are needed to keep them motivated and happy. For instance, if they feel respected and valued by the manager and by their patients, they tend to accept their tasks more easily, resulting in improved well-being at work.

Diverse studies showed that intrinsic motivation and identified regulation led to positive results for the organization, such as productivity and satisfaction, than introjected and external regulation [25]. However, this is not consensual, as other studies showed that only intrinsic motivation was significantly related to performance and well-being [26].

Motivation has been studied in light of the job characteristics model (JCM) [8] because one of the factors that most affects motivation is the characteristics of a job [27]. The JCM proposes that motivation is related to the characteristics of the tasks, namely identity, variety, meaning, autonomy, and feedback, which in turn are associated with three critical psychological states (perceived relevance, responsibility, knowledge of the results achieved at work) that lead to certain results, such as performance.

Task identity refers to a worker's ability to complete tasks and with which s/he identifies. Hackman and Oldham [8] stated that jobs that involve an intact task, such as providing a complete unit of service or putting together an entire product, are invariably more interesting to perform than jobs that involve only small parts of the task. Plus, task meaning refers to the impact of work on workers' daily life, both in and outside the organization. Hence, it is related to the degree of importance attributed to the tasks by the individuals and, as a consequence, influences their well-being [8]. Task variety refers to the need of using the abilities, knowledge, and skills to carry out the tasks. According to Sims, et al. [28], work becomes more interesting and enjoyable when it involves a set of different tasks. Autonomy has been the most studied [29]; it is the degree of freedom and the level of independence that a worker has to perform the tasks. Hence, it allows workers to become independent and able to self-manage their work, with the ability to make decisions and choose which methodologies to use to carry out the tasks [1]. That is, autonomy focuses on the individual's freedom to design work, make decisions, and define their working methods [7]. Lastly, feedback is the degree of direct and concise information about a worker's effectiveness on a task [8].

These characteristics, also known as work design [7], give rise to critical psychological states: perceived relevance, that is, the importance that the individual attributes to work according to their values, needs, and expectations, which is determined by the variety, identity, and meaning of tasks; responsibility, which is the commitment to the work results, where autonomy plays a key role; and the knowledge of the results achieved at work, that is, the awareness of self-efficiency while performing the tasks [1]. The interaction between job characteristics and critical psychological states leads to outcomes or impacts, such as intrinsic motivation, job satisfaction, individual development, decreased job turnover, and job performance [8]. These characteristics are motivating once they lead to greater performance and well-being. Therefore, based on the empirical literature, we hypothesized the following:

**Hypothesis 1 (H1).** *The relationship between job characteristics and happiness is mediated by intrinsic motivation (but not by extrinsic motivation).*

### 2.2. The Mediating Role of Adaptive Performance

Some studies have shown that low levels of autonomy and feedback decrease motivation [30]. However, this is not consensual, as some authors have argued that greater responsibility, freedom, and recognition seem to increase motivation [31].

Other authors have studied the role of job characteristics for different outputs, such as adaptive performance—the ability to adapt to certain job conditions, situations, or events [29]. This relationship occurs since there is a positive effect on the workers' psychological state, such as the intrinsic motivation to perform tasks, the responsibility for the results, and the knowledge of the achieved goals [32]. The adaptability and identity of the task and the common goals of the organization develop collaboration between workers and encourage mutual support [27]. Thus, the greater the identity of the tasks, the greater the balance between individual and organizational interests, increasing adaptive performance [33]. Thus, based on the JCS, we defined the following hypothesis:

**Hypothesis 2 (H2).** *The relationship between job characteristics and happiness is mediated by adaptive performance.*

*2.3. The Serial Mediating Role of Intrinsic Motivation and Adaptive Performance*

Nevertheless, some barriers may appear during task performance, such as lack of materials, scarce information, or workload—characteristics from healthcare working settings—which may negatively influence performance [34]. Some constraints that result from the performance of tasks, at certain times, such as stress from dealing with patients, the pressure exerted on individuals, the long working hours, and the complexity of the tasks, restrict the performance of the task; however, these constraints may encourage adaptive performance behaviors when individuals are intrinsically motivated. Hence, they are motivated to manage the constraints that limit their performance [34].

We argue that intrinsic motivation is positively related to performance adaptive behaviors and individuals' happiness—the individual's judgment about the quality of life; however, we expect that extrinsic motivation, in the form of integrated regulation, does not serve as a process that explains how certain job motivating characteristics improve happiness and performance. Integrated regulation is a form of extrinsic motivation that needs external elements to drive individuals' actions. Job characteristics such as autonomy or task meaning are intrinsically motivating due to their subjective nature. That is, the need for autonomy varies from individual to individual. Moreover, perceived autonomy is also a process of cognitive judgment of autonomy per se. Thus, combining cognitive judgment with the need for such characteristics depends on the individual. Therefore, we may not judge these job characteristics as extrinsically motivated. Based on the JCM and the SDT, we defined the following:

**Hypothesis 3 (H3).** *The relationship between job characteristics and happiness is serially mediated by intrinsic motivation and adaptive performance (but not by extrinsic motivation) (Figure 1).*

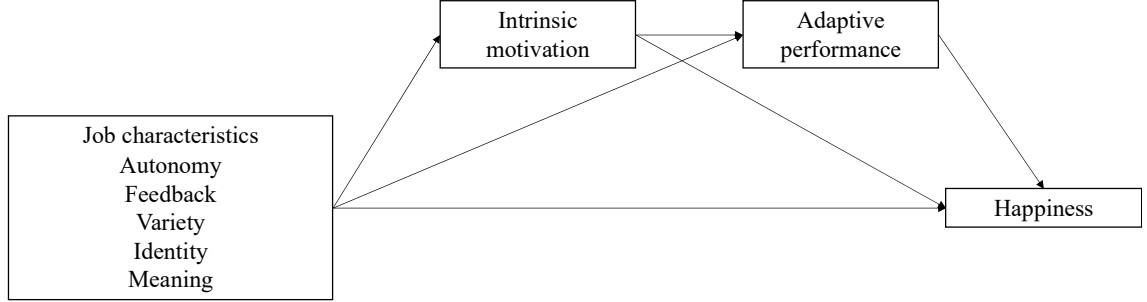

**Figure 1.** The serial mediation model.

## 3. Method

*3.1. Participants and Procedure*

We collected data with a convenience sample that included 290 healthcare workers from a nursing home. Their mean age was 40 years (SD = 12.35), and the majority were aged between 46 and 64 years old (40.2%), of which 89.6% were female and had a graduate degree (48.1%). The mean organizational tenure was 4 years (SD = 8.49), and the mean function tenure was 5.2 years (SD = 8.25). The sample comprised nurses (43%), assistants (31%), doctors (16%), and technicians (10%). On average, they worked 40 h per week (SD = 16.95).

We contacted the Board Director from the nursing home, in November 2020, to conduct the study there. We explained our main goals and assured the confidentiality and anonymity of the data. Then, we contacted the workers from that nursing home via email. In the email, we assured that the participation would be voluntary, and we clarified this study's goals and the data collection procedure. After they signed an informed consent, we sent them three emails. The first email included a link to the first survey that aimed to

measure job characteristics and the sample's socio-characterization. Two days later, we sent the second email, containing another link for another survey. This measured motivation. At last, two days later, we sent the last email for the survey. This assessed adaptive performance and happiness. Data collection took place in January 2021. From the first to the third day of data collection, 10 participants quit. Moreover, from the 300 workers contacted, we obtained 290 valid answers (response rate: 97%). The sample size is suitable to test our hypotheses because we collected data from (almost) the entire population of this nursing home (97% response rate) [35].

### 3.2. Measures

Motivation. We used the Multidimensional Work Motivation Scale [25]. This included 18 items divided into six dimensions: amotivation (e.g., "I do not exert myself at work because I think it is a waste of time"), social external regulation (e.g., "I exert myself at work because I will get more respect from others"), material external regulation (e.g., "I exert myself at work because I will only be rewarded financially by others if I try"), introjected regulation (e.g., "I exert myself at work because in exerting myself, I feel proud of myself"), identified regulation (e.g., "I think it is important to exert myself in my work"), and intrinsic motivation (e.g., "I think it's important to put effort into my work") The items were answered using a 7-point Likert scale (1 "Not at all"; 7 "Completely"). The internal consistency of the scale, measured through Cronbach's alpha, ranged between 0.77 and 0.95 (see Appendix A).

Job characteristics. We used 15 items from the Work Design Questionnaire [7]. The scale measured: autonomy (e.g., "My job allows me to plan how I do my work"), variety (e.g., "My job involves a wide variety of tasks"), task meaning (e.g., "The results of my work are likely to significantly affect the lives of other people"), task identity (e.g., "My job gives me the possibility to finish the parts of the work that I have started"), feedback (e.g., "My job itself provides feedback on my performance"). The individuals responded on a 5-point Likert scale (1—Strongly Disagree; 5—Strongly Agree). Cronbach's alphas ranged from 0.88 to 0.95.

Adaptive performance. We used three items from the scale of performance developed by Griffin., et al. [36]. A sample item is "Adapted well to changes in core tasks"). They had to respond based on a 5-point Likert-type scale (1—Very Little; 5—To a great extent). Cronbach's alpha was 0.96.

Happiness. We used the 5-item Satisfaction with Life Scale [37]. A sample item is (I am satisfied with my life). Participants answered the items on a 5-point Likert scale (1—strongly disagree, 5—completely agree). Cronbach's alpha was 0.86.

Control variables. We used sex, age, habilitations, and years of service as control variables, because these variables have been shown to account for differences in motivation, performance, and happiness indicators [38].

### 3.3. Data Analyses

We used SPSS to analyze the data and the macro-PROCESS to test our hypotheses (model 6—serial mediation analyses) [39]. PROCESS is an observed-variable modeling tool that relies on ordinary least squares (OLS). This macro is suitable as it enables the isolation of each mediator's indirect effect (intrinsic motivation (H1) and adaptive performance (H2)), as well as testing of the serial indirect effect (H3) [39]. Moreover, an added value of this macro is that it tests the indirect effect between the predictor and the criterion variables through the mediator via the bootstrapping approach, thereby addressing some flaws associated with the Sobel test [39]. This approach allows us to calculate not only the coefficients (β) and the SE (Standardized Estimates), but also the 95% CI (confidence intervals) for the path estimates. Hayes and Scharkow [40] suggested that bootstrap confidence intervals are a good approach for detecting path coefficients. Moreover, whereas other statistical programs (e.g., AMOS) provide bootstrap confidence intervals for overall indirect effects, it is not possible to obtain it for specific indirect effects [41] as PROCESS

macro does, in serial mediations. Moreover, this macro has been recognized as an added value because it allows testing even with smaller samples [42]. Hence, the PROCESS macro is suitable to test the hypotheses in this study.

To test for common method bias, we performed confirmatory factor analyses. The results show that the nine-factor model (autonomy, variety, feedback, meaning, identity, intrinsic and extrinsic motivation, adaptive performance, and happiness) fitted the data well (RMSEA = 0.07, CFI = 0.93 TLI = 0.92, SRMR = 0.05). The single-factor model evidenced an unacceptable fit (RMSEA = 0.13, CFI = 0.72 TLI = 0.69, SRMR = 0.15).

## 4. Results

Table 1 shows the means, standard deviations, and correlations for all the variables.

**Table 1.** Descriptive statistics, correlations, and Cronbach's alphas.

| Variable | M | SD | 1 | 2 | 3 | 4 | 5 | 6 | 7 | 8 | 9 |
|---|---|---|---|---|---|---|---|---|---|---|---|
| 1. Autonomy | 3.76 [1] | 1.16 | (0.88) | | | | | | | | |
| 2. Feedback | 4.13 [1] | 0.81 | 0.30 ** | (0.89) | | | | | | | |
| 3. Variety | 4.21 [1] | 0.91 | 0.46 ** | 0.37 ** | (0.90) | | | | | | |
| 4. Identity | 3.93 [1] | 0.94 | 0.35 ** | 0.57 ** | 0.26 ** | (0.88) | | | | | |
| 5. Meaning | 3.92 [1] | 1.11 | 0.33 ** | 0.27 ** | 0.42 ** | 0.31 ** | (0.90) | | | | |
| 6. Extrinsic motivation | 6.28 [2] | 1.14 | 0. 21 ** | 0.11 | 0.29 ** | 0.13 * | 0.15 * | (0.95) | | | |
| 7. Intrinsic motivation | 5.35 [2] | 1.50 | 0.41 ** | 0.30 ** | 0.37 ** | 0.29 ** | 0.32 ** | 0.54 ** | (0.95) | | |
| 8. Performance | 4.38 [1] | 0.67 | 0.28 ** | 0.31 * | 0.40 ** | 0.22 ** | 0.16 ** | 0.26 ** | 0.28 ** | (0.96) | |
| 9. Happiness | 3.66 [1] | 0.83 | 0.45 ** | 0.36 ** | 0.29 ** | 0.33 ** | 0.19 ** | 0.20 ** | 0.49 ** | 0.44 ** | (0.86) |
| Age | 40 | 12.35 | 0.09 | 0.06 | 0.05 | 0.07 | 0.14 * | −0.10 | 0.06 | 0.03 | 0.04 |
| Sex | - | - | −0.16 * | 0.10 | −0.14 * | 0.12 * | −0.13 * | −0.35 ** | −0.19 ** | −0.06 | 0.03 |
| Habilitations | - | - | −0.00 | −0.01 | −0.02 | 0.00 | −0.04 | −0.20 ** | −0.18 ** | 0.03 | 0.04 |
| Tenure | 4 | 8.49 | −0.00 | −0.01 | −0.02 | −0.05 | 0.09 | 0.00 | 0.07 | 0.08 | 0.02 |

*Note.* $N = 290$. [1] Scale from 1 to 5. [2] Scale from 1 to 7. * $p < 0.05$, ** $p < 0.01$.

Table 2 shows the indirect effects and the 95% bias-corrected bootstrapped confidence intervals (CI) for the path estimates. Since neither sex nor age statistically influenced the variables of the study, we opted to remove them from the analyses.

Hypothesis 1 stated that intrinsic motivation would mediate the relationship between job characteristics and happiness. The findings show that all the job characteristics led to happiness via intrinsic motivation (but not via extrinsic) (autonomy: β = 0.09, SE = 0.02, 95% CI [0.05, 0.14], feedback: β = 0.11, SE = 0.03, CI 95% [0.06, 0.18], variety: β = 0.13, SE = 0.03, 95% CI [0.08, 0.19], meaning: β = 0.10, SE = 0.02, CI 95% [0.06, 0.15], and identity: β = 0.09, SE = 0.03, 95% CI [0.05, 0.14]), lending support to H1.

Hypothesis 2 expected that adaptive performance would mediate the link between job characteristics and happiness. The indirect effect of adaptive performance was significant for all the job characteristics as predictors of happiness (autonomy: β = 0.04, SE = 0.02, 95% CI [0.01, 0.09], feedback: β = 0.08, SE = 0.03, 95% CI [0.03, 0.12], variety: β = 0.11, SE = 0.03, 95% CI [0.05, 0.17], meaning: β = 0.02, SE = 0.02, 95% CI [0.00, 0.06], and identity: β = 0.04, SE = 0.02, 95% CI [0.01, 0.09]). Thus, H2 was supported.

Hypothesis 3 expected that intrinsic motivation and adaptive performance serially mediated the relationship between job characteristics and happiness. The indirect effect of both intrinsic motivation and adaptive performance was significant (autonomy: β = 0.02, SE = 0.01, 95% CI [0.01, 0.04], feedback: β = 0.02, SE = 0.01, 95% CI [0.01, 0.04], variety: β = 0.02, SE = 0.01, 95% CI [0.00, 0.04], meaning: β = 0.02, SE = 0.01, 95% CI [0.01, 0.04], and identity: β = 0.02, SE = 0.01, 95% CI [0.01, 0.04]). The serial mediation was not significant for extrinsic motivation (autonomy: β = 0.01, SE = 0.00, CI 95% [−0.00, 0.03], feedback: β = 0.01, SE = 0.00, 95% CI [−0.01, 0.03], variety: β = 0.01, SE = 0.01, 95% CI [−0.00, 0.05], meaning: β = 0.01, SE = 0.01, 95% CI [−0.00, 0.03], and identity: β = 0.01, SE = 0.00, 95% CI [−0.00, 0.03]). Thus, H3 received support (see, Figure 2).

**Table 2.** Serial mediation model results.

| | Intrisic Motivation | | Adaptive Performance | | Happiness | |
|---|---|---|---|---|---|---|
| Predictor | *Coeff* | *t* | *Coeff* | *t* | *Coeff* | *t* |
| Autonomy | 0.53 ** | 7.51 | 0.11 ** | 3.13 | 0.17 ** | 4.44 |
| Intrisic Motivation | - | - | 0.09 ** | 3.22 | 0.17 ** | 5.78 |
| Performance | - | - | - | - | 0.36 ** | 5.76 |
| Age | 0.00 | 1.10 | −0.00 | −0.25 | −0.00 | −0.26 |
| Sex | −0.47 * | −2.53 | −0.03 | −0.15 | 0.50 * | 2.33 |
| Habilitations | −0.23 | −1.39 | 0.06 | 0.94 | 0.02 | 0.21 |
| Tenure | 0.03 | 0.85 | 0.02 | 1.26 | 0.00 | 0.17 |
| Indirect effect | | | | | | |
| Aut → Mot → Hap | | | 0.09 ** CI 95% [0.05, 0.14] | | | |
| Aut → Perf → Hap | | | 0.04 ** CI 95% [0.01, 0.09] | | | |
| Aut → Mot → Perf → Hap | | | 0.02 ** CI 95% [0.01, 0.04] | | | |
| $R^2$ | 17.02 | | 11.14 | | 38.32 | |
| $F$ | 56.38 ** | | 17.17 ** | | 56.52 ** | |
| Feedback | 0.57 ** | 5.35 | 0.21 ** | 4.26 | 0.18 ** | 3.26 |
| Intrisic Motivation | - | - | 0.09 ** | 3.47 | 0.20 ** | 6.80 |
| Performance | - | - | - | - | 0.36 ** | 5.57 |
| Age | −0.00 | −0.08 | −0.00 | −0.37 | −0.00 | −0.35 |
| Sex | −0.90 * | −2.04 | −0.12 | −0.62 | 0.38 | 1.68 |
| Habilitations | −0.13 | −0.81 | 0.10 | 1.52 | 0.06 | 0.69 |
| Tenure | 0.00 | 0.10 | 0.01 | 0.69 | −0.01 | −0.44 |
| Indirect effect | | | | | | |
| Fee → Mot → Hap | | | 0.11 CI 95% [0.06, 0.18] | | | |
| Fee → Perf → Hap | | | 0.08 CI 95% [0.03, 0.12] | | | |
| Fee → Mot → Perf → Hap | | | 0.02 CI 95% [0.01, 0.04] | | | |
| $R^2$ | 9.43 | | 13.67 | | 36.35 | |
| $F$ | 28.64 ** | | 21.70 ** | | 51.97 ** | |
| Variety | 0.60 ** | 6.56 | 0.26 ** | 6.01 | 0.02 | 0.33 |
| Intrisic Motivation | - | - | 0.07 ** | 2.60 | 0.22 ** | 7.24 |
| Performance | - | - | - | - | 0.40 ** | 5.97 |
| Age | −0.00 | −0.02 | −0.00 | −0.28 | −0.00 | −0.32 |
| Sex | −0.56 | −1.24 | −0.00 | −0.02 | 0.50 * | 2.20 |
| Habilitations | −0.22 | −1.29 | 0.06 | 0.07 | 0.04 | 0.49 |
| Tenure | 0.00 | 0.09 | 0.00 | 0.38 | −0.00 | −0.36 |
| Indirect effect | | | | | | |
| Var → Mot → Hap | | | 0.13 95% CI [0.08, 0.19] | | | |
| Var → Perf → Hap | | | 0.11 95% CI [0.05, 0.17] | | | |
| Var → Mot → Perf → Hap | | | 0.02 95% CI [0.00, 0.04] | | | |
| $R^2$ | 13.53 | | 18.69 | | 33.89 | |
| $F$ | 43.03 ** | | 31.48 ** | | 46.64 ** | |
| Identity | 0.45 ** | 4.97 | 0.11 ** | 2.50 | 0.15 ** | 3.29 |
| Intrisic Motivation | - | - | 0.11 ** | 3.99 | 0.20 ** | 6.79 |
| Performance | - | - | - | - | 0.38 ** | 6.03 |
| Age | −0.00 | −0.29 | −0.00 | −0.38 | −0.00 | −0.44 |
| Sex | −0.95 * | −2.15 | −0.05 | −0.25 | 0.38 | 1.67 |
| Habilitations | −0.12 | −0.73 | 0.10 | 1.45 | 0.06 | 0.66 |
| Tenure | 0.03 | 0.85 | 0.01 | 0.90 | −0.00 | −0.05 |
| Indirect effect | | | | | | |
| Id → Mot → Hap | | | 0.09 95% CI [0.05, 0.14] | | | |
| Id → Perf → Hap | | | 0.04 95% CI [0.01, 0.09] | | | |
| Id → Mot → Perf → Hap | | | 0.02 95% CI [0.01, 0.04] | | | |
| $R^2$ | 8.26 | | 10.01 | | | |
| $F$ | 24.75 ** | | 15.24 ** | | | |
| Meaning | 0.44 ** | 5.72 | 0.05 | 1.44 | 0.00 | 0.10 |
| Intrisic Motivation | - | - | 0.12 ** | 4.14 | 0.22 ** | 7.35 |
| Performance | - | - | - | - | 0.41 ** | 6.45 |
| Age | −0.00 | −0.32 | −0.00 | −0.38 | −0.00 | −0.14 |
| Sex | −0.58 | −1.23 | −0.03 | −0.14 | 0.48 * | 2.09 |
| Habilitations | −0.15 | −0.89 | 0.11 | 1.47 | 0.05 | 0.58 |
| Tenure | 0.01 | 0.35 | 0.01 | 0.84 | −0.00 | −0.25 |
| Indirect effect | | | | | | |
| Mean → Mot → Hap | | | 0.10 95% CI [0.06, 0.15] | | | |
| Mean → Perf → Hap | | | 0.02 95% CI [0.00, 0.06] | | | |
| Mean → Mot → Perf → Hap | | | 0.02 95% CI [0.01, 0.04] | | | |
| $R^2$ | 10.66 | | 8.65 | | 33.86 | |
| $F$ | 32.80 ** | | 12.98 ** | | 46.59 ** | |

Note. $N = 290$; * $p < 0.05$ ** $p < 0.001$.

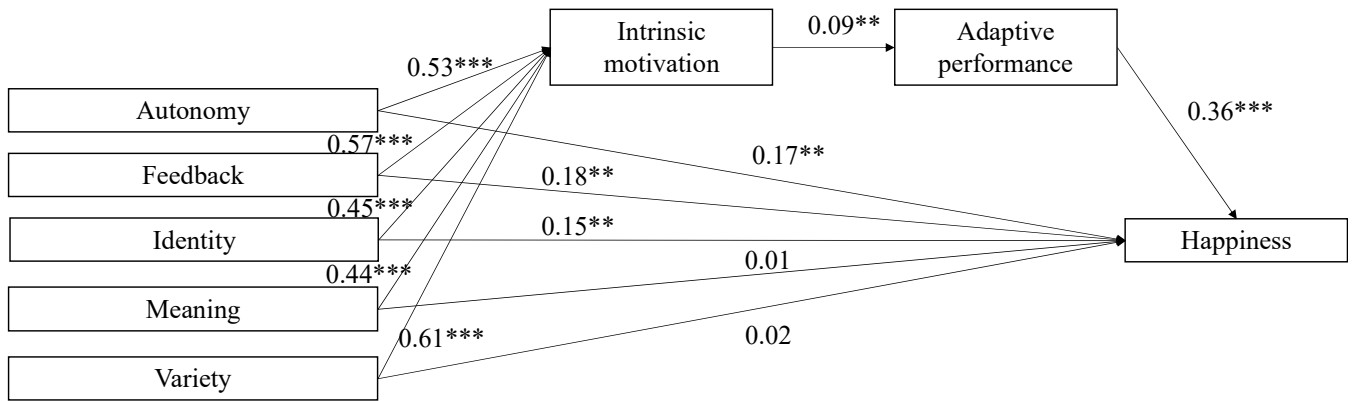

**Figure 2.** The analyzed serial mediation model (** *p* < 0.01; *** *p* < 0.001).

## 5. Discussion

The present study analyzes the relationship between five job characteristics (autonomy, feedback, variety, meaning, and task identity) and happiness via two types of motivation (extrinsic and intrinsic) and adaptive performance, in a sample of nursing home professionals that included doctors, nurses, technicians, and assistants. The findings are of particular importance given the COVID-19 crisis, which was a trigger of uncertainty [37].

The findings show that job characteristics influence healthcare workers' happiness via two processes: affective (motivation) and behavioral (adaptive performance). That is, in line with the JCS of Hakman and Oldham [1,8], we show that autonomy, feedback, task variety, identity, and meaning are indeed motivating job characteristics that positively influence intrinsic motivation, that leads to more adaptive performance behaviors, making the individual feel better and happier. When the levels of these job characteristics are higher, workers' intrinsic motivation increases, making them more committed and engaged with work, leading to higher adaptive performance, and, as a result, happiness increases. These results are consistent with the theoretical [8] and empirical [34] literature. For example, Sonnentag et al. [34] showed that the same job characteristics were positively related to performance and satisfaction through intrinsic motivation. The JCM argues that these job characteristics are motivating and lead to positive outcomes because there is a positive effect of job characteristics on individuals' psychological states, such as the responsibility for the results to be achieved, or the relevance attributed by the individuals to their work and results. Accordingly, these characteristics are motivating once they lead to greater performance and well-being. Similarly, Millette and Gagné [43] showed that job characteristics were positively associated with intrinsic motivation, satisfaction, and performance. More recently, Zaman et al. [44] showed that job characteristics influenced motivation and, therefore, joy while working and performance. Therefore, we may conclude that autonomy, feedback, task variety, identity, and meaning influence healthcare workers' intrinsic motivation, which leads to more adaptive behaviors and in turn to increased happiness.

Our findings also evidence that this serial mediation does not occur for extrinsic motivation. While some authors argued that some forms of extrinsic motivation would lead to positive outcomes, such as performance [15,25], others are in line with this result [1]. Hence, we may conclude that extrinsic motivation may be relevant to some outcomes (e.g., task performance) due to its inherent external rewards (e.g., salary, compensation) [45]; however, it is not sufficient to commit the individual to go further on his/her formal responsibilities and engage in adaptive behaviors that, in turn, result in happiness. Contrary to extrinsic motivation, some job characteristics make the individual take pleasure from work, involving him/her, and improving adaptive work behaviors. In turn, these make individuals feel satisfied with themselves and with life.

We may not forget the period in which data were collected, amid the pandemic. Hence, these results highlight the role that intrinsic motivating characteristics appear to have for healthcare workers. Therefore, this study extends the job characteristics model by

demonstrating that (1) healthcare employees' happiness is affected by job characteristics (autonomy, feedback, variety, meaning, and task identity) that influence their intrinsic motivation, because these professionals, due to their demanding working day, need to feel energized by their work, to be more adaptable to such demands. Healthcare workers often have complex and demanding working days, which were particularly difficult during the pandemic [37], as they were faced with a significantly higher quantitative overload and negative affective situations such as the death of their patients. Indeed, job motivating characteristics, such as autonomy, can be a suitable strategy to energize and motivate these employees, making them more adaptable to uncertain and complex situations, which at the same time may lead them to flourish and be happier.

### 5.1. Limitations and Future Directions

The present study has some limitations that are essentially related to the sample. Although the number of participants is considered acceptable for the study, the sample size ($N = 290$) is small and, as such, limits the generalizability of the results. Moreover, we gathered data in only one institution and with a specific class of professionals (healthcare workers), which again may limit the generalizability of the data for other professional groups and institutions. Another limitation is related to the self-report of the data; that is, participants may have given "socially desirable" answers, which limits the reliability of the conclusions obtained. Moreover, even though we have collected data at three time points, we still have a cross-sectional study that may have resulted in the common method bias. However, the confirmatory factor analyses performed minimize such bias, as it indicates that the one-factor model does not fit the data well.

Future studies should use other types of designs (e.g., diaries) to test the proposed conceptual model, because motivation, performance, and happiness are dynamic constructs influenced by the context and individual [17]. Thus, a daily design would capture such dynamics among daily motivation, daily performance, and daily happiness [46]. Additionally, it would be interesting to test this model applied to telework, given that telework has a different work design than the face-to-face model of work.

### 5.2. Practical Implications

Currently, organizations face a period of instability and volatility, which is emphasized specifically for healthcare workers. As Greenberg [12] noted, healthcare workers have been required to work in highly challenging conditions during the COVID-19 pandemic and may therefore be at increased risk of experiencing mental health problems and decreased motivation. These factors are also related to changes in daily routines at work that, in turn, influence the way healthcare workers work and may impair the quality of care they would normally provide [12].

Thereby, the results focus on the need to look after these professionals, as they look after their patients. Indeed, some job characteristics influence individuals' motivation, performance, and, as a result, their happiness. The results show that intrinsic motivation is positively related to performance; that is, employees' performance is higher when they feel intrinsically motivated. This motivation seems to improve the level of commitment and encourage individuals to achieve better results for organizations [27]. Thus, from a practical point of view, healthcare institutions must apply strategies so that employees feel connected to them, that is, feel intrinsically motivated, leading to greater performance. For example, healthcare workers should be actively monitored. This type of monitoring may be of interest for managers, as there is evidence suggesting that proactively asking these kinds of workers about their motivation, internal states, and well-being can increase the take-up of care, as was found after terrorist incidents in the UK [47]. This intervention focused on monitoring professionals who could rely on the regular application of an anonymous, online self-check tool comprising a range of motivational and well-being measures giving tailored advice, such as self-help information.

Moreover, these institutions can think of incentive plans, rewards, or attractive conditions for employees (such as flexible hours, telework, a day off on the employee's birthday or children's birthday, and cash vouchers at Christmas, among others). It is also important to promote a good working environment, create conditions for a good relationship between workers, and create moments for team building. These strategies may promote commitment and motivation for the effective performance of tasks and for healthcare workers to feel happy at work.

Overall, healthcare institutions must have open communication with an effective design of functions to foster intrinsic motivation and, consequently, achieve productivity.

## 6. Conclusions

The results of this study extend previous works on job characteristics and their predictive effect on happiness. The findings show that autonomy, feedback, task variety, identity, and meaning influence healthcare workers' intrinsic motivation, which leads to more adaptive behaviors and in turn to increased happiness. In contrast, extrinsic motivation appears to be insufficient to influence healthcare workers to improve their adaptive performance, and as a result, lead them to be happier.

**Author Contributions:** Conceptualization, methodology, software, validation, formal analysis, writing—original draft preparation, writing—review, and editing: A.J.-S.; investigation, resources, data curation, visualization: C.M. All authors have read and agreed to the published version of the manuscript.

**Funding:** This research received no external funding.

**Institutional Review Board Statement:** The study was conducted by the Declaration of Helsinki and approved by the Institutional Review Board of Instituto Politécnico de Tomar (ESGT06102020 and 10/10/2020).

**Informed Consent Statement:** Informed consent was obtained from all subjects involved in the study.

**Data Availability Statement:** Data will be made available upon reasonable request.

**Conflicts of Interest:** The authors declare no conflict of interest.

## Appendix A. Survey

**Motivation: Multidimensional Work Motivation Scale**
Why do you or would you put effort into your current job?
**Amotivation**
I don't, because I really feel that I'm wasting my time at work.
I do little because I don't think this work is worth putting efforts into.
I don't know why I'm doing this job, it's pointless work.
**Extrinsic regulation—social**
To get others' approval (e.g., supervisor, colleagues, family, clients . . . ).
Because others will respect me more (e.g., supervisor, colleagues, family, clients . . . ).
To avoid being criticized by others (e.g., supervisor, colleagues, family, clients . . . ).
**Extrinsic regulation—material**
Because others will reward me financially only if I put enough effort in my job (e.g., employer, supervisor . . . ).
Because others offer me greater job security if I put enough effort in my job (e.g., employer, supervisor . . . ).
Because I risk losing my job if I don't put enough effort in it.
**Introjected regulation**
Because I have to prove to myself that I can.
Because it makes me feel proud of myself.
Because otherwise I will feel ashamed of myself.
Because otherwise I will feel bad about myself.
**Identified regulation**

Because I personally consider it important to put efforts in this job.
Because putting efforts in this job aligns with my personal values.
Because putting efforts in this job has personal significance to me.

**Intrinsic motivation**
Because I have fun doing my job.
Because what I do in my work is exciting.
Because the work I do is interesting.
scale: 1 = "not at all", 2 = "very little", 3 = "a little", 4 = "moderately", 5 = "strongly", 6 = "very strongly", 7 = "completely"

**Job characteristics: Work Design Questionnaire**
Think about your job and indicate whether you agree or disagree with the following affirmations.

**Autonomy**

1.  My job allows me to make my own decisions about how to schedule my work.
2.  My job allows me to decide on the order in which things are done on the job.
3.  My job allows me to plan how I do my work.

**Task variety**

1.  My job involves a great deal of task variety.
2.  My job involves doing a number of different things.
3.  My job requires the performance of a wide range of tasks.

**Task meaning**

1.  The results of my work are likely to significantly affect the lives of other people.
2.  The job itself is very significant and important in the broader scheme of things.
3.  The job has a large impact on people outside the organization.

**Task identity**

1.  My job involves completing a piece of work that has an obvious beginning and end.
2.  My job is arranged so that I can do an entire piece of work from beginning to end.
3.  My job gives me the possibility to finish the parts of the work that I have started.

**Feedback from job**

1.  My work activities themselves provide direct and clear information about the effectiveness (e.g., quality and quantity) of my job performance.
2.  My job itself provides feedback on my performance.
3.  My job itself provides me with information about my performance.

    Scale (1) Strongly Disagree; (5) Strongly Agree.

**Adaptive performance**

On the last day (you worked), identify the extent to which the following items match your behavior.

1.  Adapted well to changes in core tasks
2.  Coped with changes to the way you have to do your core tasks.
3.  Learned new skills to help you adapt to changes in your core tasks

    Scale (1) very little (5) To a great extent.

**Happiness—Satisfaction with Life Scale**

Please consider the following affirmations. Indicate whether you agree/disagree with each one.

1.  In most ways my life is close to my ideal.
2.  The conditions of my life are excellent.
3.  I am satisfied with my life.
4.  So far, I have gotten the important things I want in life.
5.  If I could live my life over, I would change almost nothing.

    (1) completely disagree to (5) completely agree.

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
