# Peer review of "How Job Characteristics Influence Healthcare Workers’ Happiness: A Serial Mediation Path Based on Autonomous Motivation and Adaptive Performance"

_sustainability, doi:10.3390/su142114251_

Round 1
Reviewer 1 Report
This paper intends to investigate how job characteristics influence health care workers’ happiness and implied that the care workers’ intrinsic motivation can be enhanced and lead to better adaptive performances. However, the reviewer has some major concerns as follow:
1) The author should highlight the main contributions compared with the prior works. The literature review is not sufficient.
2) The research question should be mentioned following the introduction part. Then those questions should be fully discussed following data analysis part.
3) For the analysis results, the simple size is relatively small and should be demonstrated that it is sufficient for analysis. Why did not use Structural Equation Modeling tool (e.g., AMOS ) to analyze the collected data? More parameters need to be added, for example, R square, f square.
4) The table 2 should be reorganized.
5) The authors should make more discussion in the background of COVID-19 and figure out its influencing for healthcare workers.
Author Response
Reviewer 1
Comments
This paper intends to investigate how job characteristics influence health care workers’ happiness and implied that the care workers’ intrinsic motivation can be enhanced and lead to better adaptive performances. However, the reviewer has some major concerns as follow:
**Thank you very much for your insightful comments and suggestions. We respond to each of your specific comments in detail below. We highlight our responses with **.
1) The author should highlight the main contributions compared with the prior works. The literature review is not sufficient.
**Thank you for your comment. We added the main contributions of the paper at the end of the introduction (lines 63-79).
“The main contribution of this study is to address each of these issues. First, we extend the job characteristics model by proposing that (1) healthcare employee’s happiness may be affected by job characteristics (autonomy, feedback, variety, meaning, and task identity) that influence their intrinsic motivation because these professionals, due to their de-manding working day, need to feel energized by their work, in order to be more adaptable to such demands.
Hence, relying on the job characteristics model, we proposed a model consisting of a motivational and behavioral path, based on a serial mediation – motivation and performance. The main research question was: what job characteristics would influence healthcare employees’ intrinsic motivation, that in turn, make them more adaptable and results in increased happiness? Therefore, we aimed to analyze the path from job characteristics (autonomy, feedback, variety, meaning, and task identity) to happiness, via autonomous versus controlled motivation, and performance among healthcare workers.”
2) The research question should be mentioned following the introduction part. Then those questions should be fully discussed following data analysis part.
**As we described before, we added the research question in the introduction – “The main research question was: what job characteristics would influence healthcare employees’ intrinsic motivation, that in turn, make them more adaptable and results in increased happiness? Therefore, we aimed to analyze the path from job characteristics (autonomy, feedback, variety, meaning, and task identity) to happiness, via autonomous versus controlled motivation, and performance among healthcare workers.”
3) For the analysis results, the simple size is relatively small and should be demonstrated that it is sufficient for analysis. Why did not use Structural Equation Modeling tool (e.g., AMOS ) to analyze the collected data? More parameters need to be added, for example, R square, f square.
**We added the R-square and F in table 2. Indeed, the sample is small (we highlight it in the limitations section; however, the sample is sufficient because we had 41 items in the survey which multiplied by 5 (Singh & Masuku, 2014) gives an overall needed sample of 205. Hence, 290 workers from an institution are a sufficient sample. Plus, using the sample size calculator, from a universe of 300 healthcare workers, we would need 166 workers in the study, and we have 290. Moreover, we performed the serial mediation via SPSS because the PROCESS macro allows for a robust estimation of the model and may be compared to SEM in AMOS or R.
4) The table 2 should be reorganized.
**We have reduced the size of the letter; it is now on two sheets. I understand your concern, however, the option would be to create a table for each mediation. Do you think it should be better?
5) The authors should make more discussion in the background of COVID-19 and figure out its influencing for healthcare workers.
**We appreciate your effort in making us deep further our discussion. Indeed, we improved it, as you suggested (lines 394-402).
“We may not forget the period in which data was collected, amid the pandemic. Hence, these results highlight the role that intrinsic motivating characteristics appear to have for healthcare workers. Healthcare workers often have complex and demanding working days that were particularly difficult during the pandemic days [37], as they were faced with a significantly higher quantitative overload and negative affective situations like the death of their patients. Indeed, job motivating characteristics, such as autonomy, can be a suitable strategy to energize and motivate these employees, making them more adaptable to uncertain and complex situations which at the same time may lead them to flourish and be happier.”
**Thank you once again for your thoughtful comments and suggestions to help us improve our manuscript, in particular the theoretical framework and discussion section. We look forward to your feedback on our revisions.
Reviewer 2 Report
Dear authors,
I read your article. The topic analyzed in this study is of interest to both researchers and health practitioners.
To improve the scientific quality of your article, my recommendations are
- in the introduction, I recommend adding novelty of your study and the structure of this article
- in the literature part, there is an imbalance between the size of the 3 analyzed subchapters. In this sense, I recommend to achieve a balance regarding the literature, that supports the hypotheses included in this study
- to the research methodology part, I recommend adding tables with obtained results. Moreover, it is useful to add an appendix with the research instrument used in this study. The authors do not explain how they created the research instrument - is it their own questionnaire or is it a questionnaire tested in other researches?
- in my opinion, research limits can be a problem for your study, because this study analyzes the data collected from a single asylum, which is unrepresentative. However, may you have scientific arguments to support your sample?
- on the results side, I think that the two subchapters can be joined
- the authors omitted the part of conclusions
I hope these recommendations will be useful in improving your article!
Best wishes!
Author Response
Reviewer 2
Comments:
Dear authors,
I read your article. The topic analyzed in this study is of interest to both researchers and health practitioners.
**Thank you very much for your insightful comments and suggestions. We respond to each of your specific comments in detail below. We highlight our responses with **.
To improve the scientific quality of your article, my recommendations are
- in the introduction, I recommend adding novelty of your study and the structure of this article
**Thank you for your comment. We added the main contributions of the paper at the end of the introduction (lines 63-79).
“The main contribution of this study is to address each of these issues. First, we extend the job characteristics model by proposing that (1) healthcare employee’s happiness may be affected by job characteristics (autonomy, feedback, variety, meaning, and task identity) that influence their intrinsic motivation because these professionals, due to their demanding working day, need to feel energized by their work, in order to be more adaptable to such demands.
Hence, relying on the job characteristics model, we proposed a model consisting of a motivational and behavioral path, based on a serial mediation – motivation and performance. The main research question was: what job characteristics would influence healthcare employees’ intrinsic motivation, that in turn, make them more adaptable and results in increased happiness? Therefore, we aimed to analyze the path from job characteristics (autonomy, feedback, variety, meaning, and task identity) to happiness, via autonomous versus controlled motivation, and performance among healthcare workers.
This paper is divided into five sections. First, we describe the main theoretical assumptions that support the hypotheses and the serial mediation model. Next, we describe the methods section, followed by the hypotheses testing and its results. We finish with the results’ discussion in the light of the theoretical and former empirical evidence.”
- in the literature part, there is an imbalance between the size of the 3 analyzed subchapters. In this sense, I recommend to achieve a balance regarding the literature, that supports the hypotheses included in this study
**We appreciate your comment. Accordingly, we made some changes to the theoretical framework.
- to the research methodology part, I recommend adding tables with obtained results. Moreover, it is useful to add an appendix with the research instrument used in this study. The authors do not explain how they created the research instrument - is it their own questionnaire or is it a questionnaire tested in other researches?
**We used validated scales as it is described in the methods section (lines 258-288). We also added an appendix with such measures.
- in my opinion, research limits can be a problem for your study, because this study analyzes the data collected from a single asylum, which is unrepresentative. However, may you have scientific arguments to support your sample?
**We collected data from one healthcare institution because it was a nonprobabilistic sample methodology by convenience. We added this information in the methods section, as well as we describe this limitation in the limitations section as well. “The present study has some limitations that are essentially related to the sample. Although the number of participants is considered acceptable for the study, the sample size (N=290) is small and, as such, limits the generalizability of the results. Moreover, we gathered data in only one institution and with a specific class of professionals (health-care workers), which again may limit the generalizability of the data for other professional groups and institutions.”
- on the results side, I think that the two subchapters can be joined
**As you suggested, we joined the two subchapters.
- the authors omitted the part of conclusions
**Thank you for your comment. As you suggested, we added a conclusions section (see as follows).
“6. Conclusion
The results of this study extend previous works on job characteristics and their predictive effect on happiness. The findings show that autonomy, feedback, task variety, identity, and meaning influence healthcare workers’ intrinsic motivation, which leads to more adaptive behaviors and in turn to increased happiness. In contrast, extrinsic motivation appears to be insufficient to influence healthcare workers to improve their adaptive performance, and as a result, lead them happier.”
I hope these recommendations will be useful in improving your article!
Best wishes!
**Thank you once again for your thoughtful comments and suggestions to help us improve our manuscript, in particular the theoretical framework section. We look forward to your feedback on our revisions.
Reviewer 3 Report
Dear authors,
The manuscript is very interesting, but some correction is needed. Manuscript lacks a clearly formulated scientific problem or problematic questions. Once the scientific problem has been formulated, the results obtained will need to be adjusted accordingly. It is not provided why exactly you choose this nursing home for the research. One should think about how to present the research data in Table 2 differently. Now the data is presented on three sheets, so it is difficult to trace which graphs were on the first sheet. I would recommend that you still write the conclusions. In the practical implications text presented now, part of the text would even be suitable for the discussion part. I would like to see the main results of your work related to the scientific problem (which I also missed).
sincerely
Author Response
Reviewer 3
Comments:
Dear authors,
The manuscript is very interesting, but some correction is needed.
**Thank you very much for your insightful comments and suggestions. We respond to each of your specific comments in detail below. We highlight our responses with **.
- Manuscript lacks a clearly formulated scientific problem or problematic questions. Once the scientific problem has been formulated, the results obtained will need to be adjusted accordingly.
** We added the research question in the introduction – “The main research question was: what job characteristics would influence healthcare employees’ intrinsic motivation, that in turn, make them more adaptable and results in increased happiness? Therefore, we aimed to analyze the path from job characteristics (autonomy, feedback, variety, meaning, and task identity) to happiness, via autonomous versus controlled motivation, and performance among healthcare workers.”
- It is not provided why exactly you choose this nursing home for the research. One should think about how to present the research data in Table 2 differently. Now the data is presented on three sheets, so it is difficult to trace which graphs were on the first sheet.
**We opted by a nursing home as a convenience sample. We added that information on the paper, line 214 (method section: “We collected data with a convenience sample that included 290 healthcare workers from a nursing home.”). We have reduced the size of the letter; it is now on two sheets. I understand your concern, however, the option would be to create a table for each mediation. Do you think it should be better?
- I would recommend that you still write the conclusions.
**Thank you for your comment. As you suggested, we added a conclusions section (see as follows).
“6. Conclusion
The results of this study extend previous works on job characteristics and their predictive effect on happiness. The findings show that autonomy, feedback, task variety, identity, and meaning influence healthcare workers’ intrinsic motivation, which leads to more adaptive behaviors and in turn to increased happiness. In contrast, extrinsic motivation appears to be insufficient to influence healthcare workers to improve their adaptive performance, and as a result, lead them happier.”
- In the practical implications text presented now, part of the text would even be suitable for the discussion part. I would like to see the main results of your work related to the scientific problem (which I also missed).
**We improved the discussion section – lines 359-405.
sincerely
**Thank you once again for your thoughtful comments and suggestions to help us improve our manuscript. We look forward to your feedback on our revisions.
Round 2
Reviewer 1 Report
The reviewers' comcern had been addressed. Please consider to accept the current version.
Author Response
Thank you for your feedback which was extremely useful for the paper's quality.